# Explainable Machine Learning Framework for Image Classification Problems: Case Study on Glioma Cancer Prediction

**DOI:** 10.3390/jimaging6060037

**Published:** 2020-05-28

**Authors:** Emmanuel Pintelas, Meletis Liaskos, Ioannis E. Livieris, Sotiris Kotsiantis, Panagiotis Pintelas

**Affiliations:** 1Department of Mathematics, University of Patras, GR 265-00 Patras, Greece; livieris@upatras.gr (I.E.L.); sotos@math.upatras.gr (S.K.); ppintelas@gmail.com (P.P.); 2Department of Biomedical Engineering, University of West Attica, GR 122-43 Egaleo Athens, Greece; melletis@hotmail.com

**Keywords:** interpretable/explainable machine learning, image classification, image processing, machine learning models, white box, black box, cancer prediction

## Abstract

Image classification is a very popular machine learning domain in which deep convolutional neural networks have mainly emerged on such applications. These networks manage to achieve remarkable performance in terms of prediction accuracy but they are considered as black box models since they lack the ability to interpret their inner working mechanism and explain the main reasoning of their predictions. There is a variety of real world tasks, such as medical applications, in which interpretability and explainability play a significant role. Making decisions on critical issues such as cancer prediction utilizing black box models in order to achieve high prediction accuracy but without provision for any sort of explanation for its prediction, accuracy cannot be considered as sufficient and ethnically acceptable. Reasoning and explanation is essential in order to trust these models and support such critical predictions. Nevertheless, the definition and the validation of the quality of a prediction model’s explanation can be considered in general extremely subjective and unclear. In this work, an accurate and interpretable machine learning framework is proposed, for image classification problems able to make high quality explanations. For this task, it is developed a feature extraction and explanation extraction framework, proposing also three basic general conditions which validate the quality of any model’s prediction explanation for any application domain. The feature extraction framework will extract and create transparent and meaningful high level features for images, while the explanation extraction framework will be responsible for creating good explanations relying on these extracted features and the prediction model’s inner function with respect to the proposed conditions. As a case study application, brain tumor magnetic resonance images were utilized for predicting glioma cancer. Our results demonstrate the efficiency of the proposed model since it managed to achieve sufficient prediction accuracy being also interpretable and explainable in simple human terms.

## 1. Introduction

Image classification is a very popular machine learning domain in which Convolutional Neural Networks (CNNs) [1] have been successfully applied on wide range of image classification problems. These networks are able to filter out noise and extract useful information from the initial images’ pixel representation and use it as input for the final prediction model. CNN-based models are able to achieve remarkable prediction performance although in general they need very large number of input instances. Nevertheless, this model’s great limitation and drawback is that it is almost totally unable to interpret and explain its predictions, since its inner workings and its prediction function is not transparent due to its high complexity mechanism [2]. 

In recent days, interpretability/explainability in machine learning domain has become a significant issue, since much of real-world problems require reasoning and explanation on predictions, while it is also essential to understand the model’s prediction mechanism in order to trust it and make decisions on critical issues [3,4]. The European Union General Data Protection Regulation (GDPR) which was enacted in 2016 and took effect in 2018, demanded a “right to explanation” for decisions performed by automated and artificial intelligent algorithmic systems. This new regulation promotes to develop algorithmic frameworks which will ensure an explanation for every Machine Learning decision while this demand will be legally mandated by the GDPR. The term right to explanation [5] refers to the explanation that an algorithm must give, especially on decisions which affect human individual rights and critical issues. For example, a person who applies for a bank loan and was not approved may ask for an explanation which could be “The bank loan was rejected because you are underage. You need to be over 18 years old in order to apply.” It is obvious that there could be plenty of other reasons for his rejection, however the explanation has to be short and comprehensive [6], presenting the most significant reason for his rejection, while it would be very helpful if the explanation provides also a fast solution for this individual in order to counter the rejection decision.

Explainability can also assist in building efficient machine learning models and secures that they are reliable in practice [6]. For example, let assume that a model classified an image as a “cat” followed by explanations like “because there is a tree in image”. In this scenario, the model associated the whole image as a cat based on a tree which is indeed pictured in the image, but the explanation is obviously based on an incorrect feature (the tree). Thus, explainability revealed some hidden weaknesses that this model may have even if its testing prediction accuracy is accidentally very high. An explainable model can reveal the significant features which affect a prediction. Subsequently, humans can then determine if these features are actually correct, based on their domain knowledge, in order to make the model reliable and generalize well in every new instance in real world applications. For example, a possible “correct” feature which would prove that the model makes reliable predictions could be “this image is classified as a cat because the model identified sharp contours” this explanation would reflect that the model associated the cat with its nails which is probably a unique and correct feature that represents a cat.

Medical applications such as cancer prediction, is another example where explainability is essential since it is considered a critical and a “life or death” prediction problem, in which high forecasting accuracy and interpretation are two equally essential and significant tasks to achieve. However, this is generally a very difficult task, since there is a “trade-off” between interpretation and accuracy [4]. Imaging (Radiology) tests for cancer is a medical area in which radiologists try to identify signs of cancer utilizing imaging tests, sending forms of energy such as magnetic fields, in order to take pictures of the inner human body. A radiologist is a doctor who specializes in imaging analysis techniques and is authorized to interpret images of these tests and write a report of his/her findings. This report finally is sent to the patient’s doctor while a copy of this report is sent to the patient records. CNNs have proved that they are almost as accurate as these specialists on predicting cancer from images. However, reasoning and explanation of their predictions is one of their greatest limitations in contrast to the radiologists which are able (and obligated) to analyze, interpret and explain their decision based on the features they managed to identify in an image. For example, an explanation/reasoning of a diagnosis/prediction of a case image could probably be: “This image probably is classified as cancer because the tumor area is large, its texture color is white followed by high density and irregular shape.”

Developing an accurate and interpretable model at the same time is a very challenging task as typically there is a trade-off between interpretation and accuracy [7]. High accuracy often requires developing complicated black box models while interpretation requires developing simple and less complicated models, which are often less accurate. Deep neural networks are some examples of powerful, in terms of accuracy, prediction models but they are totally non-interpretable (black box models), while decision trees and logistic regression are some classic examples of interpretable models (white box models) which are usually not as accurate. In general, interpretability methods can be divided into two main categories, intrinsic and post-hoc [6]. Intrinsic methods are considered the prediction models which are by nature interpretable, such as all the white box models like decision trees and linear models, while post-hoc methods utilize secondary models in order to explain the predictions of a black box model. 

Local Interpretable Model-agnostic Explanations (LIME) [8], One-variable-at-a-Time approach [9], counterfactual explanations [10] and SHapley Additive exPlanations (SHAP) [6] are some state of the art examples of post-hoc interpretable models. Grad-CAM [11] is a very popular post-hoc explanation technique applied on CNNs models making them more transparent. This algorithm aims to interpret any CNN model by “returning” the most significant pixels which contributed to the final prediction via a visual heatmap of each image. Nevertheless, explainability properties such as fidelity, stability, trust and representativeness of explanations (some essential properties which define an explanation as “good explanation”) constitute some of the main issues and problems of post hoc methods in contrast to intrinsic models. Our proposed prediction framework is intrinsic model able to provide high quality explanations (good explanations).

In a recent work, Pintelas et al. [7], proposed an intrinsic interpretable Grey-Box ensemble model exploiting black-box model’s accuracy and white-box model’s explainability. The main objective was the enlargement of a small initial labeled dataset via a large pool of unlabeled data adding black-box’s most confident predictions. Then, a decision tree model (intrinsic interpretable model) was trained with the final augmented dataset while the prediction and explanation were performed by the white box model. However, one basic limitation is that the application of a decision tree classifier or any white box classifier on raw image data without a robust feature extraction framework would be totally inefficient since it would require an enormous amount of images in order to build an accurate and robust image classification model. In addition, the interpretation of this tree would be too complicated since the explanations would rely on individual pixels. Our proposed prediction framework would be able to provide stable/robust and accurate predictions followed by explanations based on meaningful high-level features extracted from every image.

This paper proposes an explainable machine learning prediction framework for image classification problems. In short, it is composed by a feature extraction and an explanation extraction framework followed by some proposed “conditions” which aim to validate the quality of any model’s predictions explanations for every application domain. The feature extraction framework is based on traditional image processing tools and provides a transparent and well-defined high level feature representation input, meaningful in human terms, while these features will be used for training a simple white box model. These extracted features aim to describe specific properties found in an image based on texture and contour analysis. The feature explanation framework aims to provide good explanations for every individual prediction by exploiting the white box model’s inner function with respect to the extracted features and our defined conditions. It is worth mentioning that the proposed framework is general and can potentially be applied to any image classification task. However, this work aims to apply this framework on tasks where interpretation and explainability are vitally and significantly prominent. To this end, it was chosen to perform a case study application on Magnetic Resonance Imaging (MRI) for brain cancer prediction. In particular, we aim to diagnose and interpret glioma, which is a very dangerous type of tumor, being in most cases a malignant cancer [12], versus other tumor types, which are most of the times benign. Some examples of extracted meta-features for this case study are tumor’s size, tumor’s shape, texture irregularity level and tumor’s color.

The contribution of this work lies on the development of an accurate and robust prediction framework, being also intrinsic interpretable, able to make high quality explanations and make reasoning and justification on its predictions for image classification tasks. For this task, it is proposed a feature extraction framework which creates transparent and meaningful high level features from images and an explanation extraction framework which exploits a linear model’s inner function in order to make good explanations. Furthermore, we propose and define some conditions which aim to validate the quality of any model’s explanations on every application domain. In particular, if one model verifies all these conditions, then its predictions’ explanations can be considered as good. Finally, 3 types of presentation forms are also proposed for the prediction’s explanations with respect to the target audience and the application domain.

The remainder of this paper is organized as follows. Section 2 presents in a very detailed way our proposed research framework while Section 3 presents our experimental results, regarding the prediction accuracy and model interpretation/explanation. In Section 4, a brief discussion regarding our proposed framework is conducted. Finally, Section 5 sketches our conclusive remarks and possible future research.

## 2. Materials and Methods

### 2.1. Proposed Explainable Prediction Framework

Figure 1 presents the abstract architecture of our model which is composed by the feature extraction framework, which will be described in next subsection, a white box linear model and an explanation extraction framework. The great advantage of lineal models is that the identification of the most important features is very easy since it naturally comes out by the interpretation of their corresponding weights. A low weight’s absolute value indicates low importance while in contrast a high, indicates high importance. We have to mention that although linear models are considered intrinsic interpretable models, this does not imply that they can also provide by default good explanations. Therefore, it was developed a new explanation extraction framework that will exploit the inner linear model’s prediction function in such a way that will provide an easily understandable explanation output scheme which will satisfy all the proposed conditions that verify an explanation as good.

### 2.2. Image Feature Extraction Methodology

In general, the input feature representation which a machine learning model uses to make predictions or the feature representation input that an explanation algorithm uses to make explanations (e.g., post hoc models) must be understandable to humans when the objective is the development of an explainable prediction model [8]. Therefore, we introduce a feature extraction framework which will create transparent and well-defined features from images by processing the pixel values from grayscale images based on traditional image analysis techniques, while these features will be also applied as an input on a prediction model. This feature extraction step is essential in order to convert the initial noisy pixel representation input of every image to a compressed, compact, robust and meaningful high-level feature representation input for the final prediction model while this representation will be meaningful to humans too. Following such approach, we can get rid of the black box approach of utilizing CNN models for image classification tasks and use instead simple, stable and intrinsic interpretable models as final predictors such as linear models. 

It is worth mentioning that probably some of the utilized features will not be totally understandable to every human since the main audience domain which are directed to is for image analysts and therefore this fact points out our main approach’s limitation. Based on this limitation, in Section 2.2.4 we provide a qualitative explanation in human understandable terms for some of these complex features in order to expand our model’s explanations audience range. However, easily defined features such as size of object (e.g., tumor size in our case study), mean value of image pixels (e.g., tumor color tends to be black or white) and cyclic level (e.g., tumor shape tends to be cyclic or oval) are by default easily understandable to most humans but they may not be significant for the prediction problem which a machine learning model aims to solve. Therefore, identifying features understandable to humans being also informative and useful for the machine learning model, constitute a key factor for creating efficient and viable explainable prediction models.

#### 2.2.1. Data Acquisition

As mentioned before, the main objective of this work is to apply the proposed framework on tasks where interpretation and explainability is vital and very significant. Therefore, we chose to perform a case study application on MRI for glioma tumor classification. The utilized dataset is publicly available [13]. In particular, it is consisted of 3064 head images with glioma (1426 slices), meningioma (708 slices) and pituitary (930 slices). The slices consist of 512×512 resolution, $0.49\times 0.49\;{\mathrm{mm}}^{2}$ voxel size and tumor depicted in three planes (axial, coronal and sagittal). The main task of this case study is the identification of glioma tumor, since in most cases, the glioma lesion tends to be malignant while meningioma and pituitary (the other two main types of brain tumor) tend to be benign. Therefore, the images were separated in two groups. The first group includes glioma with 1426 instances and the other meningioma and pituitary with 1638 instances. 

Our feature extraction procedure consists of two main image feature family types, texture and contour features. Texture features are extracted based on the image’s pixel values intensities (e.g., gray levels) while contour features based on the shape of the Region of Interest (ROI) lied into the image. The final extracted dataset consists of 234 different features in total composed by 194 texture features and 40 contour features. Some examples of these extracted features from each image are area size of tumor, pixels mean value, intensity, variation, correlation, smoothness, coarseness and regularity [14]. 

The first step constitutes the identification of the ROI. The ROI was evaluated quantitatively using the manual segmentation by experts as ground truth. Figure 2 and Figure 3 present some examples of extracted tumor areas. 

The body of tumor is represented by pixels with different gray levels. Every pixel has significant information about the type of tumor. Furthermore, we normalized the images from 0 to 32 gray levels. The low values represent the dark color pixels and the high values the light white pixels.

#### 2.2.2. Texture Features 

Texture analysis is one of the high-speed feature extraction methods for image analysis and classification using the pixel’s values. These methods mainly aim to describe the spatial distribution of intensities and the varying shades of pixels in images. In this study, three major approaches were utilized, the gray-level Co-Occurrence matrix, Run length and Statistical values from ROI matrix. 

Co-Occurrence matrix [14] characterizes the texture of an image by capturing the gray-level values with spatial dependence from image, calculating how often pairs of pixel with specific values and in a specified spatial relationship occur in an image. In particular, we used the initial ROI matrix of every image and four more scales from wavelet transform [15] as pre-processing step, in order to find all pixel information. Then, the Co-occurrence matrices for every ROI image (initial plus four extracted filtered images) were calculated. The Co-Occurrence matrix is calculated by the following equation:$$G=\left[\begin{array}{cccc} p\left(1,1\right) & p\left(1,2\right) & \cdots & p\left(1,N_{g}\right)\\ p\left(2,1\right) & p\left(2,2\right) & \cdots & p\left(2,N_{g}\right)\\ \vdots & \vdots & \ddots & \vdots \\ p\left(N_{g},1\right) & p\left(N_{g},2\right) & \cdots & p\left(N_{g},N_{g}\right) \end{array}\right]$$

$N_{g}$ is the number of gray levels in the image and p(i,j) is the probability that a pixel with value *I* will be found adjacent to a pixel of value *j*. Based on the Co-Occurrence matrix, meaningful features for every image can be extracted such as entropy, energy, contrast, homogeneity, variance and correlation. In total, we extracted 100 features via the Co-Occurrence approach. Some of the most significant identified features are presented in Table 1.

Where  x and y are the coordinates (row and column) of the entry matrix, $p_{x+y}\left(i\right)$ is the probability of coordinated summing to x+y, $\mu_{x}$, $\mu_{y}$, $\sigma_{x}$ and $\sigma_{y}$ are the means and standard deviations of $p_{x}$ and $p_{y}$ which are the partial probability density functions,
$$HXY=-\overset{N_{g}}{\underset{i=1}{\sum }}\overset{N_{g}}{\underset{j=1}{\sum }}p\left(i,j\right)log\left\{p\left(i,j\right)\right\}\;\;HXY_{2}=-\overset{N_{g}}{\underset{i=1}{\sum }}\overset{N_{g}}{\underset{j=1}{\sum }}p_{x}\left(i)p_{y}(j\right)log\left\{p_{x}\left(i)p_{y}(j\right)\right\}\;$$
and  HX, HY are the entropies of $p_{x}$ and $p_{y}$.

Run-length matrix is another way to characterize the surface of a given object or region utilizing a *gray level run*. A gray level run is a set of consecutive image points computed in any direction [16]. The matrix element (i,j) specifies the number of times that the pixel value appears in every direction. These elements correspond to the number of homogeneous runs of j voxel with gray value i [17]. From run-length method were produced 39 features. The most important run-length features identified via the interpretation of the weights of our utilized white box linear prediction model are Short Run Emphasis (SRE), Run Length Non-Uniformity Normalized (RLNUN) and Low Gray Level Run Emphasis (LGLRE) as presented in Table 2.

Where *Nr* is the number of different run-length that occurs. Higher value of SRE indicates fine textures. RLNU measures the distribution of runs over the gray values and LGLRE is the distribution of the low gray-level runs [16]. These feature values are low when runs are equally distributed along grey levels indicating higher similarity in intensity values [17]. 

Statistical Values are features based on first and second order statistical analysis. The texture of an image is determined by the distribution over the pixels in calculated region. These features aim to describe specific properties of an image, such as the smoothness or irregularity, homogeneity or inhomogeneous while some extracted features are mean value, standard deviation, kurtosis, entropy, correlation and contrast.

#### 2.2.3. Contour Features

In general, contour features aim to describe the characteristics and the information lied in the shape of objects. In our case study, tumor shape constitutes the object on every image as presented in Figure 4. For example, such features can describe if the tumor shape tends to be irregular or regular, oval or cyclic, large or small and so on. In our study, tumor size was identified to be the most significant contour feature (based on formulas described in Section 2.3.4).

This feature is described by the following formula: $$tumor\;size=N_{in}$$
where $N_{in}$ is the number of pixels inside the tumor object. It is worth mentioning that this is an informative feature for the final prediction model while it is also easily understandable in human terms. In general, the identification of features which are easily understandable to most humans while they are also useful for the machine learning prediction task, probably constitute some of the main research challenges and key elements in explainable machine learning domain. The identification of such useful and easily understandable features can contribute positively to this domain since the explanation of the model’s decision would rely on features understandable in human terms and thus it would be able to address a much larger audience.

#### 2.2.4. Qualitative Explanation of Extracted Features 

We recall that an explainable machine learning model requires also an explainable feature representation in which the explanation output will rely on. Therefore, in the sequel it is attempted to explain in easy human understandable terms some complicated mathematically defined features of our framework. Table 3 presents a qualitative description of some of the most important identified features. For each of those features, we present two extreme example cases (high-low value comparison) in order to illustrate their main characteristics differences.

Correlation feature measures the linear relationship between two variables, which in our case, these variables are pixel coordinates. In other words, correlation measures the similarity between neighborhood sub-image’s regions. In natural images, neighboring pixels tend to have similar values while a white-noise image exhibits zero (lowest value) correlation. A high correlation indicates the existence of an object, objects or some kind of structure lied in the image while a low correlation could indicate that there is no object in the image or it is not transparent enough. Regarding our case study, a correlation value can indicate how clear or not, the tumor can be seen in an image. For example, in Table 3, images on the 1st column (left) correspond to a very high correlation in contrast to images on the 2nd column (right). In the left image, a tumor (object) is clearly defined, in contrast to the right which no object can be seen at all. 

Information measure of correlation measures the amount of information which one variable contains about another (pixel coordinates for images). A high value can indicate that the pixel’s intensities values of an image will tend to be smooth and regular while a low value that its pixel’s intensities values will tend to irregular. This feature for our specific study can be defined also as “tumor’s texture irregularity level”.

Sum average feature aims to measure the average of the grey values from ROI. Although its mathematic definition in Table 1 seems to be a bit complicated, in practice it measures a very easily understandable in human terms feature. In particular, it aims to describe how light or dark is by average a gray image. In our case study, a high value indicates that the tumor’s color tends to be white, a low value to be black, while an average value indicates that the tumor’s color tends to be gray.

### 2.3. Machine Learning Prediction Explanations

Interpretable machine learning focuses on interpreting machine learning models’ prediction mechanism in a reasonable form. More specifically, it aims to describe the mathematic or rule formula which an algorithm utilizes to make predictions, in a compact and meaningful form. An example is presented in Figure 5. Algorithm’s interpretation mainly aims to present and interpret (e.g., via visualization of their decision boundaries) the model’s decision function behavior. This can be very useful for machine learning developers, experts and data scientists in order to understand how a prediction model works, to identify its weaknesses and further improve its performance.

Explainable machine learning aims to explain machine learning models’ predictions to human understandable terms. The definition of explainability is a very blur issue since it deals with humans and social aspects and thus explanations can be unclear and subjective [6]. Therefore, we have to define the properties of machine learning models’ prediction explanations and clarify what should be considered as a good explanation. Our analysis below is based on the recent work of Christoph Molnar in his book Interpretable machine learning [6], the popular research work of Ribeiro et al. [8] and the resent study of Robnik-Šikonja and Bohanec [9].

#### 2.3.1. Properties of Explanations

In general, an explanation usually has to relate in some way the features utilized by the algorithm in order to make predictions understandable to human terms. Some basic properties that explanations have, are described below.

Expressive Power is the language of the explanation output which is provided by the prediction framework. Some language examples are if-then rules, decision trees, weighted sums, finite state machines, graph diagrams, etc. This property is a very significant starting step for an explanation method since the way that this language will be defined, will determine the quality and the understandability of explanations to humans. Every machine learning algorithm has its own raw language in which makes decisions and predictions depending on its own unique computation formula and algorithm. If we just utilize the algorithm’s raw output as the explanation output giving it to humans, it will be impractical, confusing and most likely not understandable at all. 

For example, the raw interpretation output of a logistic regression algorithm (considered as white box model) applied on a specific dataset composed by three features could be:$$\;\mathrm{if}\;\frac{1}{1+e^{-\left(2+5x_{1}+7x_{2}-3x_{3}\right)}}>0.5\;\mathrm{then}\;\mathrm{Output}=1\;\mathrm{else}\;\mathrm{Output}=0\;$$
which is probably not understandable and not meaningful to most humans. In contrast, decision trees models have by nature expressive power very close to human general logic since their explanation output come in an if-then-else total rule form composed by an ensemble of sub-rules which relate only one feature at a time. For example, an explanation output of a decision tree could be:if {(sub-rule 1) and (sub-rule 2) and (sub-rule 3)} then (Output =1)else (Output=0)
where sub-rules 1, 2 and 3 could be $x_{1}=“\mathrm{poor}”$, $x_{2}=“\mathrm{young}”$,${\;x}_{3}>95\;\mathrm{kg}$ where $x_{3}=“\mathrm{weight}”$, respectively.

Then this is close to human reasoning and understanding as an expressive language.

Fidelity is a property, which describes how well the explanation output approximates the actual raw decision function of the prediction algorithm. For example, a decision tree model has very high fidelity since its decision function matches exactly with the utilized explanation output. In contrast, an explanation output for a black box model utilizing post-hoc methods has to approximate the original black box decision function for example with simple intrinsic models. 

However, global fidelity which describes how well the explanation output approximates the complete model’s decision function providing global explanations, is almost impossible to achieve in practice and thus the explanations focus on local fidelity which deals with specific data instances or local data instances and data sub-sets, providing instance and local explanations. Individual instances and local explanations are usually more significant in practice, since humans mostly care about explanations for every individual instance at a time or for instances, which belong to the same vicinity (similar instances), instead of giving a global detailed explanation for every possible instance which could lead to information overload and very confusing explanations. Nevertheless, global explanations or explanations which cover as much more instances and data sub-sets at the same time (also called representativeness of explanations), are very significant in order to ascertain trust in the model.

Stability refers to the ability of a prediction model to provide similar explanations for similar instances. This means that slight changes in the feature values of an instance will not substantially change the explanation. High stability is desirable since it can enhance the trust and reliability of model’s decisions and explanations. Notice that a beneficial side effect of the stability property is that it can also diagnose possible overfitting behavior of a prediction model since lack of explanations’ stability highlights also high variance of the model’s decision function. Finally, it is worth mentioning that the more complicated a model’s decision function is, the more unstable this model will probably be, while its function interpretation will be more difficult. This means that simple models with simple prediction function are stable models. However, this does not mean that all white box models are stable (since white boxes are usually considered simple models). A decision tree with a very large max depth will probably lead in a very complicated prediction function being probably extremely unstable just like a common complicated black box model.

Comprehensibility refers to the ability of an explanation to be short and comprehensive being understandable by most humans. This property is very significant since it defines how informative and understandable at the same time is an explanation. Comprehensibility is highly depended on the target audience. This means that different explanations may need for example, a mathematician/statistician comparing to the manager or director of a company. Therefore, this property is highly affected by the expressive power as mentioned before, since an intelligent choice and definition of the explanation’s expressive power can lead to a very high comprehensibility factor with respect to the audience that the specific explanation is directed to.

Degree of importance refers to the ability of an explanation to reflect the most important features which affect the predictions. These features can describe the model’s decision function in a global way, in a local way and in individual way. Global important features are the features which highly affect the decision function of the model taking into consideration all the instances trained on. Local important features are the features which highly affect the model’s decision function taking into consideration local data sub-sets, while instance’s important features are the features which highly affect the prediction just only for this specific individual instance. This global–local differentiation is very crucial since “*features that are globally important may not be important in the local context, and vice versa*” [8]. Therefore, an explanation for an individual prediction mostly cares for the specific features that affect this specific instance or similar to it instances, rather than describing the global important features.

#### 2.3.2. Fundamental Property of Explanations

Nevertheless, even if an explanation possesses all of these properties, the main fundamental property than an explanation should have, is its humanistic social property which comes from the interactive dialogue between the explainer and the audience. An explanation which covers all factors for a certain prediction is not a human friendly explanation while the explanations have to be selected and come gradually via small answers based on the questions applied. More specifically, this dialogue comes up in a “questions–answers” form, where questions are in general contrastive and answers (explanations) are selected. 

It was proved that humans’ questions usually are contrastive and humans most of the times expect and desire selected explanations [6]. The term contrastive question means that questions are in a counterfactual form, e.g., “*Why the model predicted output 1 instead of 2?*”, “*What the prediction would be if feature input X would be different?”, “What is the minimum change required for feature input X in order the model to predict a different output result?*”. The term selected means that explainers usually select only one or two causes and present these as the explanation [6]. Therefore good explanations have to be able to answer contrastive questions and be selected, while this does not necessary imply that every other property described on previous section is not significant.

#### 2.3.3. Proposed Conditions for Good Explanations 

The definition and the validation of a good explanation can be considered in general extremely subjective and unclear. Therefore, taking into consideration the above analysis, we try to define three basic conditions that any prediction algorithm has to satisfy in order its prediction explanations to be considered good, stable, useful and easily understandable. 

Condition A. Identification of features which highly determine the prediction result for the specific individual instance. This is probably the most significant and basic part of every explanation method since most of them mainly aim to identify these features. For example, the Grad-CAM method identifies and returns the pixels of an image which are important and determine a specific prediction for a black box model such as CNN. If the pixels’ location, color level and volume can be considered as the raw initial features of every image, then such methods typically return the most significant features just like linear models would do via the interpretation of their weights on every feature variable. 

Nevertheless, it is worth mentioning that by just returning the most significant pixels of an image which determined a specific prediction it does not always lead to useful explanations. If the objective is the identification of higher level representation features and properties that come out by pixels’ grouping, such as shape and texture properties of an object lying in an image, which determine a prediction output, then relying only on pixels returns as explanations cannot reveal any useful information. For example, if a doctor classified a tumor image as malignant cancer and the main reason for this decision was just the tumor size and tumor’s irregularity level, then pixels by itself are useless. Our proposed framework will be able to provide such type of explanations and this is the main difference comparing to other works which provide explanations based on image pixel returns.

However, explanations with pixels returns is very useful when the objective is the segmentation of an object (in our case tumor) or the identification of an object hidden in a noisy image, since the return of significant pixels would reveal if the model decided correctly utilizing the correct area of pixels or wrongly. For example, if a model classified an image as a cat, where this image actually illustrates a man hugging a dog and a cat, then explanations based on pixels returns would reveal if the model decided by cat’s pixels (proper area) or another area e.g., dog’s pixels where that would be obviously wrong. 

Condition B. For a specific individual prediction identify some other instances such as local data sub-sets or instances that belong to the same vicinity (similar instances) which share the same prediction output and share at least one common explanation rule. This condition deals with the stability (robustness) and the representation property of model’s explanations and as a result with the trust factor of a prediction model. An untruthful or unstable model, such as Deep Neural Network [18], probably will provide totally different explanations for similar instances. A very common example constitutes the adversarial attacks which aim to modify an original image in a way that the changes are almost undetectable to the human eye while the prediction function of an unstable model will probably be highly affected. In a very recent study, One pixel attack for fooling deep neural networks, 2019 [19] the authors revealed that the output of Deep Neural Networks can be easily altered by adding relatively small perturbations to the input pixel vector revealing that such models could be totally unstable and unreliable on image classification tasks.

Let assume a scenario in which an explanation method like Grad-CAM, identified for one new image (instance) the important pixels which determined a specific prediction. In order to ascertain some trust in the model, it would be very helpful if an explanation framework could also provide answers to questions like “*For those identified pixels, what are their volume values in which this prediction remains same*?” A possible answer (explanation) could be “If the mean value of those pixels’ volume is higher than 180 then the prediction will remain stable”. In practice this means that for every new image which shares the same important identified pixels (or at least close to it) and the same rule: mean volume value > 180, then the model’s prediction will remain the same. Obviously, if a method can provide meaningful explanations in a global way, this is desirable too, but generally this is almost impossible in practice and as already mentioned in previous section, humans usually care for individual explanations and explanations for local or similar instances. 

Condition C. Identification of the most important features’ critical values in which the prediction result will change. This condition aims to validate the explanation method’s ability of answering contrastive questions based on the social humanistic property for making questions in a counterfactual form, as presented in previous section. Following the same previously defined scenario, some possible contrastive questions could be *“If those identified pixels were changing color and volume, what the prediction would be, would it be still the same? If not, what is the minimum change required for those pixel’s mean volume value in order the model to predict a different output result?”* A possible answer (explanation) could be “If their mean volume value increases by 30 then the prediction will change.”

Final Presentation form. Merge all explanation’s information in a compact, selected, comprehensive and informative form, with respect to the targeted audience and application domain (significant the proper definition of expressive power). This step deals with the explanation framework’s ability to create comprehensive and selected explanations which constitute some of the most significant properties of good explanations as already described in previous section. For instance, if the explanation framework was a decision tree, by default a decision tree is an example which provides information in a comprehensive and easy to read form, since one can just follow its nodes and easily verify and obtain the prediction result [7]. Nevertheless, we have to mention that even a decision tree which is naturally interpretable and explainable model, has to provide selected explanations in the case that the tree’s maximum depth is very large. This means that one has to prune and select explanation rules from the global tree in order to provide selected and comprehensive individual explanations. This is a significant limitation because this procedure is not straightforward since it must define a probably subjective threshold for this pruning procedure.

#### 2.3.4. Explanation Extraction Framework

Let assume the Logistic Regression (LR) [20] algorithm our utilized linear prediction model (it is worth mentioning that this framework theoretically can be applied on every linear interpretable prediction model). We have to note that one basic limitation of a Logistic Regression model is that it is restricted for binary classification problems and thus this limitation applies also to the proposed total prediction framework.

The prediction function of the trained LR model is given by the following formula:(1)$$f_{LR}=\{\begin{array}{cc} 1 & F>0.5\\ “unidentified“ & F=0.5\\ 0 & F<0.5 \end{array}$$
where F is defined by the following function:(2)$$F=\frac{1}{1+e^{-\left(a_{0}+\sum_{j=1}^{N}a_{j}x_{j}\right)}}$$
N is the total number of features, while if F=0.5 then the prediction output $f_{LR}$ will be “unidentified” meaning that the output can be either 0 or 1. In such cases, the prediction output can be chosen randomly or set by default to one value. By solving the inequality in Equation (1), the initial function can be simplified to: (3)$$f_{LR}=\{\begin{array}{cc} 1 & G>0\\ “unidentified” & G=0\\ 0 & G<0 \end{array}$$
where G is defined by the following function:(4)$$G=a_{0}+\overset{N}{\underset{j=1}{\sum }}a_{j}x_{j}$$
Let assume a new instance:$$I_{i}=\left(x_{i1},\;x_{i2},\ldots ,x_{ij},\ldots x_{iN}\right)$$
Verifying explanation Condition A. In linear models the absolute values of weights $a_{j}$ of each feature $x_{j}$ express the importance factors of its prediction function. A high weight value of $a_{j}$ for feature $x_{j}$ indicates that this feature is important because small changes of feature $x_{j}$ multiplied by a large $a_{j}$ weight value will highly affect the final output. The important features are chosen by the following formula:(5)$$Most\;important\;features:\;K=\left\{j:\;\left|a_{j}\right|>d_{th}\right\}$$
where $d_{th}$ is a defined threshold which defines the minimum feature’s importance factor.

Verifying explanation Condition C. For the features *k* identified as important: *k* ∈
*K*, we will compute their critical values in which the prediction result will change. These critical values are defined when the prediction output is in the “unidentified” region state. Based on Equations (3) and (4) the critical feature values $xcrit_{k}\;$ satisfy the following equation:(6)$$a_{0}+\sum_{j=1}^{k-1}a_{j}x_{ij}+a_{k}xcrit_{ik}+\sum_{j=k+1}^{N}a_{j}x_{ij}=0,\;\;\forall \;k\;\in \;K$$
and it turns out that:(7)$$xcrit_{ik}=-\frac{a_{0}+\sum_{j=1}^{k-1}a_{j}x_{ij}+\sum_{j=k+1}^{N}a_{j}x_{ij}}{a_{k}},\;\;\;\;\;\forall \;k\;\in \;K$$

Verifying explanation Condition B. If $x_{k}<xcrit_{ik}$ by Equation (7) it turns out that:$$a_{0}+\overset{k-1}{\underset{j=1}{\sum }}a_{j}x_{ij}+a_{k}x_{k}+\overset{N}{\underset{j=k+1}{\sum }}a_{j}x_{ij}<0\;$$
while if $x_{k}>xcrit_{ik}$ similarly it turns out that:$$a_{0}+\overset{k-1}{\underset{j=1}{\sum }}a_{j}x_{ij}+a_{k}x_{k}+\overset{N}{\underset{j=k+1}{\sum }}a_{j}x_{ij}>0\;$$
and thus by Equations (3) and (4) it turns out that:(8)$$y=\{\begin{array}{c} 1,\;\;\;\forall \;x_{k}\geq \;xcrit_{ik}\;and\;x_{j}=x_{ij},\;\forall \;j\neq k\\ 0,\;\;\;\forall \;x_{k}<xcrit_{ik}\;and\;x_{j}=x_{ij},\;\forall \;j\neq k \end{array}$$
where y is the prediction output. For sake of simplicity we defined by default the prediction output to be 1 for the “undentifined” state. 

Summarizing, the function described in Equations (3) and (4) represent the global interpretation formula of our prediction model while Equation (8) describes a local interpretation for instances similar to $I_{i}$ which share the same prediction output, sharing one common explanation rule. 

Final Presentation form. We have to find out a comprehensive and understandable form that will merge all information provided by conditions A, B and C while humans will be able by just following some basic rules, to easily obtain all this information. 2 main types of presentation forms are proposed for the explanation output of our prediction model: graph form since they can provide in an intelligible and compact way explanations to humans and a question–answers form since it is probably one of the most common and desirable ways that humans make explanations (Section 2.3.2). We will present it analytically in next section on our application case study scenario.

### 2.4. Summary of Proposed Framework 

In Table 4 are summarized and described in a compact form our total proposed framework’s basic steps.

## 3. Results

In this section, we present our experimental results regarding to the proposed explainable prediction framework for image classification tasks, applying it on glioma prediction from MRI as a case study application scenario. In our experiments, all utilized machine learning models (Table 5) were trained using the new data representation which was created via our feature extraction framework and validated using a 10-fold cross-validation using the performance metrics: Accuracy *(*Acc*)*, $F_{1}$-score ($F_{1})$, Sensitivity *(*sen*)*, Specificity *(*spe*)*, Positive Predictive Value *(*PPV*)*, Negative Predictive Value *(*NPV*)* and the Area Under the Curve (AUC) [21]. It was considered not essential to conduct experiments based on the initial dataset (raw images) since such experiments were already performed by various CNN models based on transfer learning approach on previous works [22,23] managing to achieve around to 99% accuracy score. It is worth mentioning that since this work proposes an explainable intrinsic prediction model, obviously our goal is not to surpass the performance of these powerful black box models but manage to achieve a decent performance score with powerful explainability. 

Our experiments were performed via two phases. In the first phase, various white box (WB) models were compared while in the second phase, the best identified WB model was compared with various black box (BB) models. Table 5 depicts all the utilized machine learning models and their basic tuning parameters. All Decision Trees (DT) models were evaluated based on their max depth parameter. A high depth leads to a complex decision function while a low depth leads to a simple function but probably to biased predictions. The basic version of decision tree algorithm used in our experiments was CART algorithm since it was identified to exhibit superior performance comparing to other decision tree algorithms [24]. On Naive Bayes (NB) [20] classifier, no parameters were specified.

All Neural Networks (NNs) are fully connected networks composed by two hidden layers each and N.L.x refers to the number of Neurons in Layer x. The basic tuning parameter of a Logistic Regression (LR) constitutes the regularization parameter C, just like in Support Vector Machine (SVM) [25], where small values specify stronger regularization. All SVM models were composed by a radial basis function kernel. We have to mention that all these models’ parameters were identified via exhaustive and thorough experiments in order to incur the best performance results. Finally, the k-NN [26] was implemented based on the Euclidean distance metric, while the basic tuning parameter is the number of neighbors k.

### 3.1. Experimental Results

Table 6 presents the performance comparison of WB models regarding the predefined performance metrics. LR_1_ exhibited the best classification score (94%) while NB exhibited the worst (77%). Table 7 presents the performance comparison of the best identified WB model (LR_1_) comparing to the BB models. LR_1_ managed to be as accurate as the best identified BB models (NN_3_ and SVM_2_). This probably means that our feature extraction framework managed to filter out the noise and the complexity of the initial image representation. As a result, this framework creates a robust and simpler data representation that simple linear models can efficiently be applied on, while powerful BB models like NN are becoming unnecessary.

### 3.2. Predictions Explanations

In the sequel, our framework’s explanation output is presented for some case study predictions. The final prediction model is the LR_1_. Let assume two new instances as presented in Figure 6.

Two basic language forms are proposed for our model’s explanation output, graph diagrams and questions–answers forms as presented in Figure 7 and Figure 8, which were extracted via the formulas described in Section 2.3.4 regarding the predefined conditions A, B and C. The graph diagram provides in a compact, comprehensive and visual form, information which can easily fast extracted by just investigating every node. Each node represents one feature followed by an explanation rule in which a specific prediction output is qualified. The three displayed nodes represent the three most important identified features while the size of each node represents the importance factor of the corresponding feature.

A questions–answers form is probably one of the best ways to provide explanations since this is the main fundamental way that humans make explanations (more details in Section 2.3.2). As already mentioned before, the proper choice of the language is highly depended by the audience. Therefore, we also propose two types of questions–answers forms, Specific and Humanistic form. In specific form, the answers are extracted directly by the graph diagram without any information loss providing all details of graph’s information. In humanistic form the answers are extracted via a preprocessing step aiming to simplify the explanation and convert it to a more human like explanation, by approximating the initial model’s features to easier understandable abstract features (meta-features) specified by the application domain. For example, in our case the Object size can be converted to Tumor Size, the Sum Average to Tumor’s Color (more details for Sum Average feature are presented in Section 2.4). Additionally, every quantitative value has to be converted to qualitative such as Small, Average, Large. For this step is essential the knowledge of a High and Low value of each feature in order to create such qualitative terms.

## 4. Discussion

In this study, a new prediction framework was proposed, which is able to provide accurate and explainable predictions on image classification tasks. Our experimental results indicate that our model is able to achieve a sufficient classification score comparing to state of the art CNN approaches being also intrinsic interpretable able to provide good explanations for every prediction/decision result.

One major difference comparing to other state of the art explanation frameworks for image classification tasks, is that our approach is not performing pixels based explanations. By the term pixels based (or pixel returns) we mean that the explanations are based on the visual interpretation of the most important identified pixels that determined a specific prediction. For example, if a model classified as a cat, an image which presents a cat and a dog, then meaningful pixel base explanation would probably return the cat’s pixels revealing that the model classified the image utilizing the proper area of pixels. However, if the task was to recognize an owner’s missing cat, then the identification of high level features which uniquely describe this cat would be essential. Such features could be cat’s size, cat’s color, cat’s color irregularity level and cat’s number of legs. In such cases, the model’s prediction explanation would be useful to rely on such high level features instead of just specific pixels. If the main reason for this prediction was just that this cat has 3 legs and blue color then pixel returns probably could not reveal any useful explanation and reasoning. 

As already mentioned our explanation approach is not pixel based but higher level feature based. By this term it is meant that the explanations are performed via a higher level feature representation input, in contrast to raw pixels which are the lowest level input (initial representation). Such high level features can describe the unique properties that groups of pixels possess in every image such as color of an image, color irregularity level, shape irregularity of objects lied in an image, number of objects lied in an image and so on. Obviously, these high-level feature inputs have to be understandable to humans since the model’s predictions’ explanations would be useless. For example, if a model classified an image as a dog because x=5 without any knowledge about what the feature x means and how was calculated, then such explanation can be considered meaningless. Where instead if it was known that the feature x is the size of an object in an image or the color value of an image and so on, then we would be able to make reasoning about the model’s decision and easily understand it.

Nevertheless, creating high level image features being also understandable to humans is a complex task. There is no guarantee that utilizing these features as an input for the machine model would lead to high prediction performance, as probably there are a lot of other hidden unutilized features lied in an image that are probably useful for the specific classification problem. It is hard to identify a priori such features since they are actually found out and crafted by a human sense based approach, while it could make more sense to seek the assistance of an expert with respect to the application domain. In contrast, automatic methods such as CNN models manage to automatically identify useful features in images avoiding this painful human based feature extraction process. However, these features that automatic methods manage to identify are not interpretable and explainable to humans, whereas features crafted by humans can be transparent, meaningful and understandable. This is the trade-off that we need to endure if the objective is the development of interpretable prediction models. Traditional feature extraction approaches, specialized expertise, specific knowledge domain regarding the application and the art of creating useful features for machine learning problems followed by new innovative strategies and techniques can constitute essential key elements in explainable artificial intelligent era. 

## 5. Conclusions and Future Work

In this work, an accurate, robust and explainable prediction framework was proposed for image classification tasks proposing three types of explanations outputs with respect to the audience domain. Comparing to most approaches, our method is intrinsic interpretable providing good explanations relying on high-level feature representation inputs, extracted by images. These features aim to describe the properties that the pixels of an image possess such as its texture irregularity level, object’s shape, size, etc. One basic limitation of our approach is that some of these features are probably only understandable by image analysts and specific human experts. However, we made an attempt to qualitatively explain what such features describe in simple human terms in order to make our model’s explanation output more attractive and viable to a much wider audience.

Last but not least, in our experiments we utilized all features, even the least significant. We attempted to reduce the number of features in this dataset by analyzing the correlation between the features as well as their significance and by applying some feature selection techniques [27,28,29]. However, any attempt of removing any features was leading to decrease the overall performance of the prediction model; hence, no feature was removed. We point out the feature selection processing was not in the scope of our work since our main objective was the development and the presentation of an explainable machine learning framework for image classification tasks. Clearly, an interpretable prediction model, exhibiting even better forecasting ability, could be developed through the imposition of sophisticated feature selection techniques as a pre-processing step.

In future work, we aim to incorporate and identify features more understandable to human, being also very informative for the machine learning prediction models. In addition, it is worth investigating whether an interpretable prediction model exhibiting even better forecasting ability could be developed through the imposition of penalty functions together with the application of feature selection techniques or through additional optimized configuration of the proposed model. Finally, we also aim to develop more sophisticated algorithmic methods in order to improve the prediction performance accuracy of intrinsic white box models. Such algorithms could simplify the initial structure complexity and nonlinearity level of the initial dataset, in order to efficiently train simple white box models.

## Figures and Tables

**Figure 1 jimaging-06-00037-f001:**
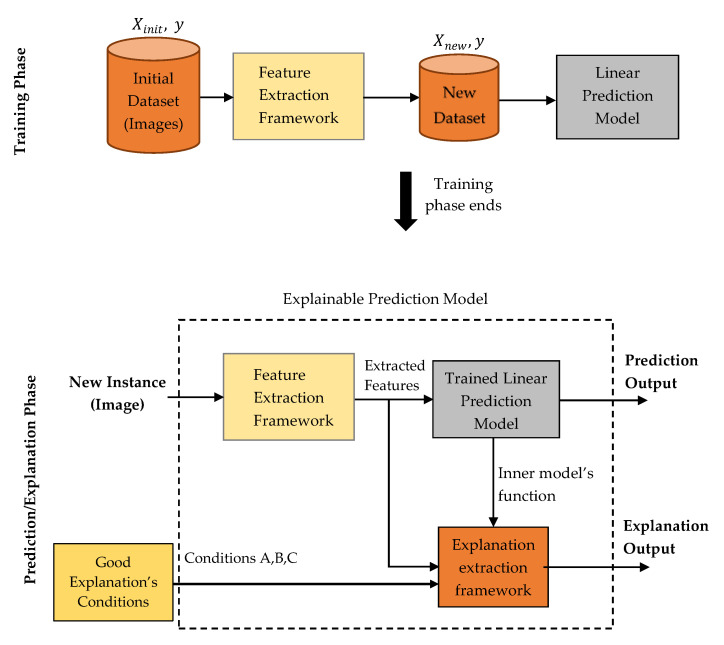
An abstract overview of our proposed explainable model.

**Figure 2 jimaging-06-00037-f002:**
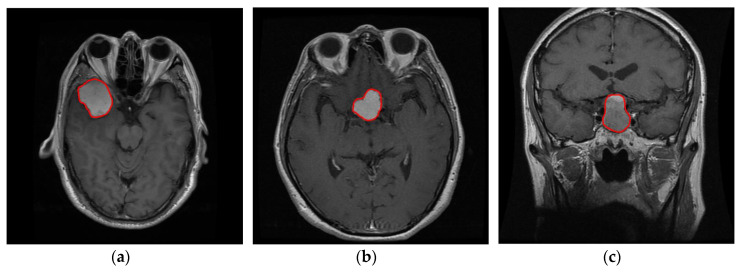
Head MRI examples. The red color illustrates the tumor area (**a**) glioma, (**b**) meningioma, (**c**) pituitary.

**Figure 3 jimaging-06-00037-f003:**
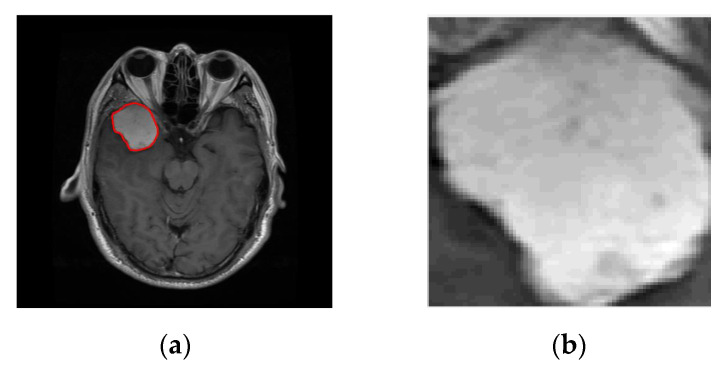
(**a**) Image with glioma, (**b**) Region of Interest (ROI) extraction.

**Figure 4 jimaging-06-00037-f004:**
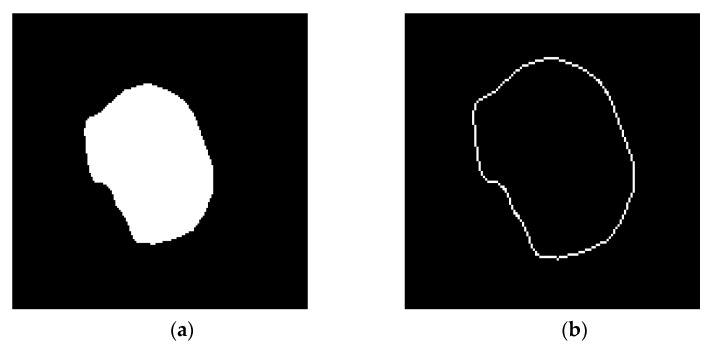
(**a**) Binary image, (**b**) border extraction.

**Figure 5 jimaging-06-00037-f005:**
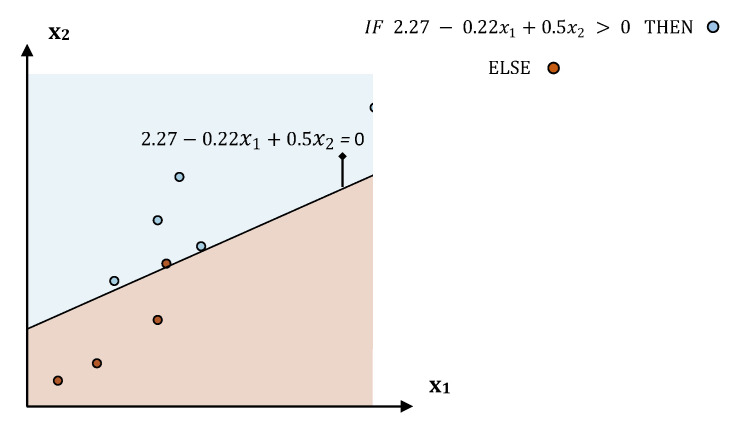
A visualization example presenting the decision function of a trained linear model.

**Figure 6 jimaging-06-00037-f006:**
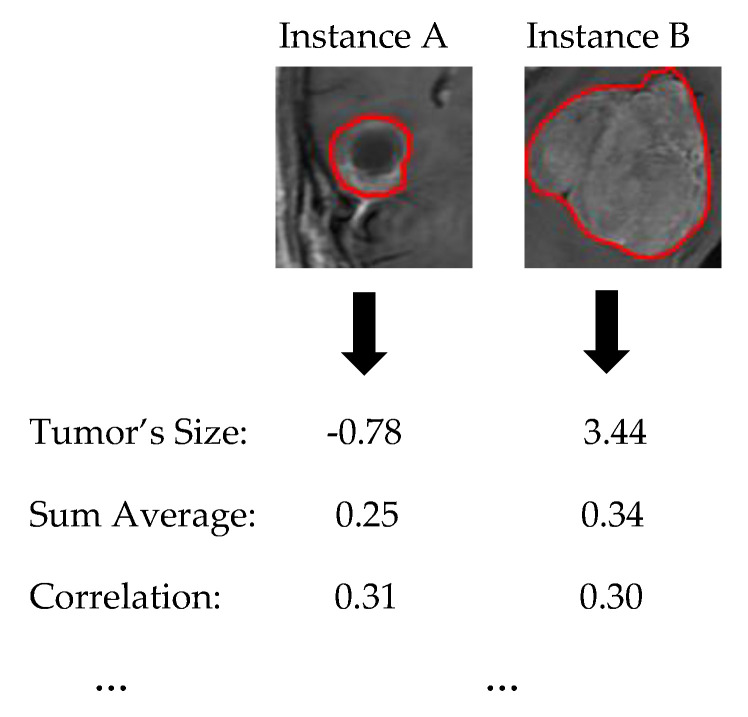
Two case study instances. Instance A is GLIOMA while Instance B is NON GLIOMA.

**Figure 7 jimaging-06-00037-f007:**
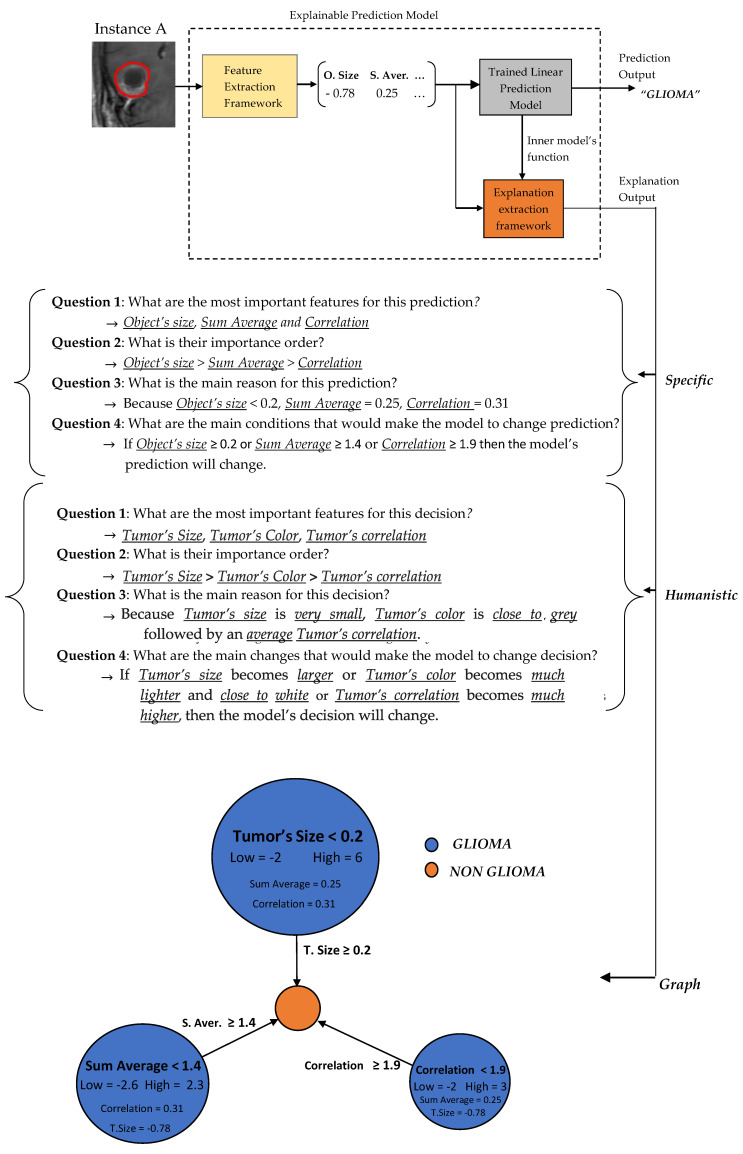
Explanation output for Instance A.

**Figure 8 jimaging-06-00037-f008:**
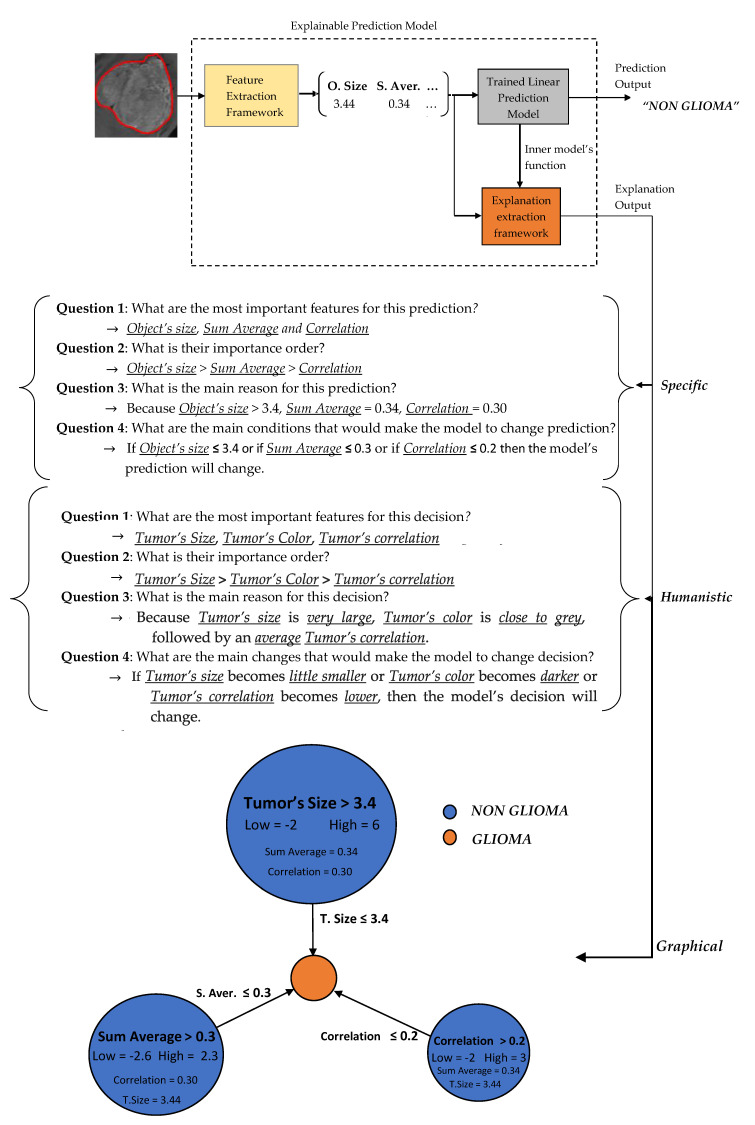
Explanation output for Instance A.

**Table 1 jimaging-06-00037-t001:** Mathematic description of the most important identified features of Co-Occurrence approach.

Co-Occurrence Features	Formula
Correlation	$\frac{1}{\sigma_{\chi }\sigma_{y}}\left(\sum_{i=1}^{N_{g}}\sum_{j=1}^{N_{g}}p\left(i,j\right)-\mu_{\chi }\mu_{y}\;\right)$
Information Measure of Correlation	${\left(1-exp\left[-2\left(HXY_{2}-HXY\right)\right]\right)}^{\frac{1}{2}}$
Sum Average	$\sum_{i=2}^{2N_{g}}ip_{x+y}\left(i\right)$

**Table 2 jimaging-06-00037-t002:** Mathematic description of the most important identified features of Run Length approach.

Run-Length Features	Formula
SRE	$\frac{1}{N}\sum_{i=1}^{N_{g}}\sum_{j=1}^{N_{r}}\frac{p\left(i,j\right)}{j^{2}}$
RLNU	$\frac{1}{N^{2}}\sum_{i=1}^{N_{g}}{\left(\sum_{j=1}^{N_{r}}p\left(i,j\right)\right)}^{2}$
LGLRE	$\frac{1}{N}\sum_{i=1}^{N_{g}}\sum_{j=1}^{N_{r}}\frac{p\left(i,j\right)}{i^{2}}$

**Table 3 jimaging-06-00037-t003:** Qualitative description of some of the most important identified features.

Features	Tumor Examples		Description
	(High Value)	(Low Value)	
Correlation	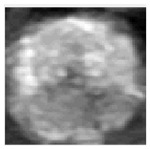	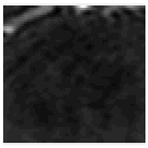	Measures the linear relationship between two variables (pixels coordinates for images). In other words, measures the similarity between neighborhood sub-image’s regions.
Information Measure of Correlation	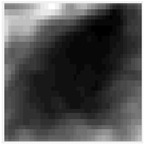	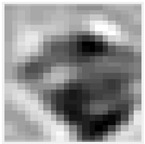	Measures the amount of information that one variable contains about another. In other words, it measures how irregular is the texture of an image.
Sum Average	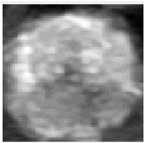	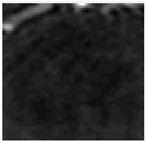	Provides the average of the grey values from ROI.

**Table 4 jimaging-06-00037-t004:** Summary of proposed framework.

**Step 1**. Import images $\left(X_{init}^{M\times H\times W},\;y\right)$, where M is the number of images, H and W are the number of pixels corresponding to the Height and Width of every image. **Step 2**. Compute Co-Occurrence, Run-length, Statistical and Contour features and extract new dataset $\left(X_{new}^{M\times N},\;y\right)$, where N is the number of new extracted features. **Step 3**. Train White Box Linear model LR with $\left(X^{M\times N},\;y\right)$. **Step 4**. Define a weight threshold $d_{th}$ and compute most important features K. **Step 7**. For every new instance $X_{new}^{1\times N}$ and every feature k ∈ K compute its critical values $xcrit_{k}$. **Step 8**. Verify explanation Conditions A, B and C. **Step 9**. Select and define the language of the explanation with respect to the targeted audience. **Step 10**. Create the Final Presentation explanation output.

**Table 5 jimaging-06-00037-t005:** Summary of utilized machine learning models.

WB Model	Basic Parameters	BB Model	Basic Parameters
DT_1_	max depth = 3	NN_1_	N.L.1=64 , N.L.2=16
DT_2_	max depth = 5	NN_2_	N.L.1=128 , N.L.2=32
DT_3_	max depth = 10	NN_3_	N.L.1=256 , N.L.2=64
NB	No parameters	SVM_1_	C=1
LR_1_	C=1	SVM_2_	C=500
LR_2_	C=500	SVM_3_	C=1000
LR_3_	C=1000	k-NN	k=3

**Table 6 jimaging-06-00037-t006:** Performance comparison of white box (WB) models.

WB Model	Acc	$\;F_{1}$	sen	spe	PPV	NPV	AUC
DT_1_	0.87	0.86	0.83	0.92	0.9	0.86	0.87
DT_2_	0.89	0.87	0.86	0.91	0.89	0.88	0.88
DT_3_	0.87	0.86	0.86	0.88	0.87	0.88	0.87
NB	0.77	0.74	0.69	0.84	0.79	0.76	0.77
LR_1_	**0.94**	**0.93**	**0.94**	**0.94**	**0.93**	**0.95**	**0.94**
LR_2_	0.93	**0.93**	0.93	0.93	0.92	0.94	0.93
LR_3_	0.93	**0.93**	0.93	0.93	0.92	0.94	0.93

**Table 7 jimaging-06-00037-t007:** Performance comparison of black box (BB)–WB models.

Model	Acc	$\;F_{1}\;$	sen	spe	PPV	NPV	AUC
NN_1_	0.93	0.92	0.92	0.94	0.93	0.93	0.93
NN_2_	0.94	0.93	0.92	0.94	0.93	0.94	0.93
NN_3_	0.94	0.93	0.92	**0.95**	**0.94**	0.93	0.94
SVM_1_	0.92	0.91	0.9	0.93	0.92	0.92	0.92
SVM_2_	0.93	0.93	0.92	**0.95**	**0.94**	0.93	0.93
SVM_3_	0.93	0.93	0.92	0.94	0.93	0.93	0.93
k-NN	0.89	0.87	0.83	0.94	0.92	0.86	0.88
LR_1_	0.94	0.93	**0.94**	0.94	0.93	**0.95**	0.94

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
