# Peer review of "Explainable Machine Learning Framework for Image Classification Problems: Case Study on Glioma Cancer Prediction"

_2313-433X, 2020, doi:10.3390/jimaging6060037_

Round 1

Reviewer 1 Report

In this paper, the authors present a new approach for developing an interpretable machine learning model for image classification problems. Their approach is interesting, however there are some issues, which should be explained/discussed.

Moreover, the authors extracted a number of features from an image. However, they did not applied any feature selection technique or study the correlation between all these features. Are all those features useful? The authors should include feature selection algorithms in their study and present their impact. Additionally, the authors should study if there exists any significant correlation between involved features.

In Line 583, the authors state “The final prediction model is the LR1.” Clearly, for a different dataset, the prediction model which reported the best performance may change. How this affects the prediction explanation analysis?

Moreover, Logistic Regression (LR) is utilised for binary classification while Decision Trees and Naïve Bayes can be used for multi-label classification as well. In the literature (including medical imaging) exists a variety of classification benchmarks with three or more classes. The authors should discuss this limitation since Section 3.2 is based on the predictions of LR.

Since the presented approach cannot perform similarly with the state-of-the-art Convolutional Neural Networks, why the use of penalty functions was not included?

Finally, Figures 6 and 7 should be removed and replaced with Tables including more performance metrics e.g. sen, spe, ppv, npv, AUC etc. This is really important, especially since we are dealing with a medical classification task.

Last but not least, the presentation should be improved.

  • There are many useless double/triple spaces observed.
  • In Figures 9 and 10, the size of blue circles should be the same as well as the size of the text.
  • Some parameters are sometimes presented using equation editor and some as a text. For example parameter C.
  • There are some typos and some sentences are too long.
  • Line 102. What does “EXPLAIN” stands for?
  • Line 284. Remove comma after “recall that”
  • F-measure metric is referred sometimes as F1 and sometimes as F1.
  • The utilised font sometimes different. See for example lines 343 and 348.

Reviewer 2 Report

This paper proposes a prediction framework which is able to provide accurate and explainable predictions on image classification tasks. Experimental results indicates that the model is able to achieve a sufficient classification score comparing to state of the art CNN approaches being also intrinsic interpretable able to provide good explanations for every prediction/decision result.

Good and interesting paper.

Do you have any Git repo with code? (if so, mention it)

Figures 6,7. Do you have color images? (not gray scale)

line 23, 25, 30, 124, 135, 136, etc. "we" ---> use impersonal form in the paper (this paper proposes, this study shows, etc..)

Round 2

Reviewer 1 Report

The authors provided a detailed response letter in respect to my comments. The paper can be now be accepted in its currect form.